# Development of a Novel Anti−CD44 Monoclonal Antibody for Multiple Applications against Esophageal Squamous Cell Carcinomas

**DOI:** 10.3390/ijms23105535

**Published:** 2022-05-16

**Authors:** Nohara Goto, Hiroyuki Suzuki, Tomohiro Tanaka, Teizo Asano, Mika K. Kaneko, Yukinari Kato

**Affiliations:** 1Department of Molecular Pharmacology, Tohoku University Graduate School of Medicine, 2-1 Seiryo-machi, Aoba-ku, Sendai 980-8575, Japan; s1930550@s.tsukuba.ac.jp; 2Ph.D. Program in Human Biology, School of Integrative and Global Majors, University of Tsukuba, Tsukuba 305-8575, Japan; 3Department of Antibody Drug Development, Tohoku University Graduate School of Medicine, 2-1 Seiryo-machi, Aoba-ku, Sendai 980-8575, Japan; tomohiro.tanaka.b5@tohoku.ac.jp (T.T.); teizo.asano.a7@tohoku.ac.jp (T.A.); k.mika@med.tohoku.ac.jp (M.K.K.)

**Keywords:** CD44, monoclonal antibody, esophageal cancer

## Abstract

CD44 is a cell surface glycoprotein, which is expressed on normal cells, and overexpressed on cancer cells. CD44 is involved in cell adhesion, migration, proliferation, survival, stemness, and chemo−resistance. Therefore, CD44 is thought to be a promising target for cancer diagnosis and therapy. In this study, we established anti−CD44 monoclonal antibodies (mAbs) by immunizing mice with a CD44 variant (CD44v3−10) ectodomain and screening using enzyme−linked immunosorbent assay. We then characterized them using flow cytometry, Western blotting, and immunohistochemistry. One of the established clones (C_44_Mab−46; IgG_1_, kappa) reacted with CD44 standard isoform (CD44s)−overexpressed Chinese hamster ovary−K1 cells (CHO/CD44s) or esophageal squamous cell carcinoma (ESCC) cell lines (KYSE70 and KYSE770). The apparent *K*_D_ of C_44_Mab−46 for CHO/CD44s, KYSE70, and KYSE770 was 1.1 × 10^−8^ M, 4.9 × 10^−8^ M, and 4.1 × 10^−8^ M, respectively. C_44_Mab−46 detected CD44s of CHO/CD44s and KYSE70, and CD44 variants of KYSE770 in Western blot analysis. Furthermore, C_44_Mab−46 strongly stained the formalin−fixed paraffin−embedded ESCC tissues in immunohistochemistry. Collectively, C_44_Mab−46 is very useful for detecting CD44 in various applications.

## 1. Introduction

CD44 is a cell surface glycoprotein, which is expressed on normal cells, and overexpressed on cancer cells. [1]. The CD44 gene is comprised of 19 exons in human [2]. The first five and the last five exons are constant and encode the shortest isoform of CD44 (85–95 kDa), called the CD44 standard isoform (CD44s). CD44s is widely distributed and plays critical roles in normal homeostasis, including hematopoiesis and immune system, and organogenesis [3]. The middle nine exons can be alternatively spliced and assembled with the ten exons contained in CD44s. They are referred to as CD44 variant isoforms (CD44v) [4]. For instance, CD44v3−10 is expressed in epithelial tissues and overexpressed in carcinomas [3]. Furthermore, CD44v3−10 functions as a co−receptor of receptor tyrosine kinases by binding to heparin−binding epidermal growth factor-like growth factor and fibroblast growth factors (via v3−encoded region) [5] and hepatocyte growth factor (via v6−encoded region) [6], and confers oxidative stress resistance via v8−10−encoded region [7]. Moreover, both CD44s and CD44v are major receptors for hyaluronic acid (HA), which is involved in cell adhesion, migration, and proliferation [8]. They also play important roles in tumor progression, metastasis, stemness, and resistance to chemo− and radiotherapy [9,10]. Therefore, CD44 is a promising target for cancer diagnosis and therapy.

Several CD44−targeting monoclonal antibodies (mAbs) have been developed for preclinical research applications. IM7 mAb was shown to inhibit HA−induced vascular endothelial growth factor production in human vascular endothelial cells [11] and significantly decrease cell migration and invasion in breast cancer cells [12]. Another mAb, H4C4 mAb, was found to reduce tumor growth, metastasis, and post−radiation recurrence [13]. A humanized mAb specific for CD44, RG7356, has been found to be directly cytotoxic for leukemia B cells, but with no effect on normal B cells. Administration of RG7356 to immune−deficient mice engrafted with human chronic lymphocytic leukemia cells was found to result in complete clearance of engrafted leukemia cells [14]. Finally, CD44v6−specific humanized mAbs (BIWA−4 and BIWA−8) labeled with ^186^Re showed the therapeutic efficacy in head and neck SCC xenograft bearing nude mice in another study [15].

Sulfasalazine is an inhibitor of xCT, the subunit of cystine–glutamate transporter. CD44v has been found to interact with the xCT and promotes oxidative stress resistant in cancer cells [7,16]. The clinical trials of sulfasalazine have been evaluated in combination with cisplatin in patients with CD44v−expressing advanced gastric cancer refractory to cisplatin [17], and the reduction of the CD44v−positive gastric cancer cells was observed in sulfasalazine treated patients [18]. These results suggest that CD44v is an important marker to predict and evaluate the sensitivity of sulfasalazine.

Previously, we established the novel mAb, C_44_Mab−5 (IgG_1_, kappa) against CD44 using the Cell−Based Immunization and Screening method [19]. C_44_Mab−5 showed high sensitivity for flow cytometry and immunohistochemical analysis in oral cancers. In this study, we developed a novel anti−CD44 mAb, C_44_Mab−46 (IgG_1_, kappa), and evaluated it for various applications, including flow cytometry, Western blotting, and immunohistochemical analyses.

## 2. Results

### 2.1. Flow Cytometric Analysis of C_44_Mab−46 to CD44 Expressing Cells

In this study, C_44_Mab−46 was established by immunizing one mouse with human CD44v3−10 ectodomain (Figure 1). We already confirmed the epitope of C_44_Mab−46 as _174_−TDDDV−_178_ [20,21] encoded within the first five exons, indicating that C_44_Mab−46 recognizes CD44 standard isoforms (CD44s). Therefore, we confirmed the reactivity of C_44_Mab−46 against CHO/CD44s by flow cytometry. As shown in Figure 2A, C_44_Mab−46 recognized CHO/CD44s cells in a dose dependent manner, but not CHO−K1 cells (Figure 2B). The recognition of CHO/CD44s by C_44_Mab−46 was strongly inhibited by CD44 peptide (amino acids 161−180, WT), which contains _174_−TDDDV−_178_ sequence. However, the D175A mutant peptide, which could not be recognized by C_44_Mab−46 [21], never inhibited (Figure 2C). Furthermore, C_44_Mab−46 also recognized endogenous CD44 in the context of ESCC cell lines as it reacted with both KYSE 70 and KYSE770 in a dose−dependent manner (Figure 2D,E).

Next, we assessed the binding affinity of C_44_Mab−46 with CHO/CD44s, KYSE70, and KYSE770 using flow cytometry. The apparent *K*_D_ of C_44_Mab−46 for CHO/CD44s, KYSE70, and KYSE770 was 1.1 × 10^−8^ M, 4.9 × 10^−8^ M, and 4.1 × 10^−8^ M, respectively, indicating that C_44_Mab−46 possesses moderate affinity for CD44s−expressing cells (Figure 3).

### 2.2. Western Blot Analysis

Western blotting was performed to further assess the sensitivity of C_44_Mab−46. Lysates of CHO−K1 and CHO/CD44s cells were probed. As shown in Figure 4A, C_44_Mab−46 detected CD44s as a ~85 kDa band. However, C_44_Mab−46 did not detect any band from lysates of CHO−K1 cells. These results indicated that C_44_Mab−46 specifically detect exogenous CD44s. Next, we examined the detection of endogenous CD44 using lysates from KYSE70 and KYSE770 cells. As shown in Figure 4B, C_44_Mab−46 could detect CD44 as 85 and 48kDa double bands from lysates of KYSE70 cells. In contrast, C_44_Mab−46 could detect CD44 as more than 100 kDa triplet bands, probably CD44v isoforms, from lysates of KYSE770 cells. We also performed the peptide blocking experiment and found that 85 kDa band in CHO/CD44s (Figure 4C), 85 and 48 kDa bands in KYSE70, and CD44v isoforms in KYSE770 (Figure 4D) were blocked in the presence of the CD44 peptide (amino acids 161−181, WT). These results suggest that C_44_Mab−46 recognizes endogenous CD44s and CD44v(s) in the esophageal cancer cell lines.

### 2.3. Immunohistochemical Analysis Using C_44_Mab−46 against ESCC Tissues

To investigate whether C_44_Mab−46 can be used for immunohistochemical analyses using paraffin−embedded tumor sections, we used tissue microarray of ESCC. In a well differentiated ESCC section (Figure 5A–F), C_44_Mab−46 strongly stained ESCC cells (Figure 5A). A clear membrane−staining in ESCC was observed (Figure 5B). Hematoxylin and eosin (HE) staining was performed using the serial sections (Figure 5E,F). In an ESCC section with stromal invaded phenotype (Figure 5G–L), C_44_Mab−46 strongly stained stromal invaded ESCC (Figure 5G,H). HE staining was also performed using the serial sections (Figure 5K,L). Additionally, C_44_Mab−46 also stained at basal layer and several layers above in normal squamous epithelium of esophagus (Appendix A). We also performed the peptide blocking experiment and found that the recognition of ESCC tissues by C_44_Mab−46 was completely blocked by CD44 peptide (amino acids 161−180, WT) (Appendix A). In immunohistochemical analyses using C_44_Mab−46 against ESCC tissue microarrays (Table 1), C_44_Mab−46 stained 63 of 67 (94.0%) cases of ESCC. These results indicated that C_44_Mab−46 is useful for immunohistochemical analysis of paraffin−embedded tumor sections.

## 3. Discussion

CD44 have functions in many processes in normal cells (hematopoietic, immune system, and organogenesis), and in pathological situations (inflammation and cancer) [3]. CD44 exhibits passive ligand binding, including adhesion to hyaluronan and other components of the extracellular matrix. Furthermore, CD44 acts as a co−receptor for growth factors and as a scaffold for enzymes. Therefore, establishment and characterization of anti−CD44 mAbs are thought to be important for the development of CD44 targeting therapy and diagnosis.

In this study, we developed C_44_Mab−46, which can recognize all isoforms of CD44, and shows utility in flow cytometry, Western blotting, and immunohistochemistry applications. As shown in Figure 2, C_44_Mab−46 similarly recognized endogenous CD44 of KYSE70 and KYSE770 cells by flow cytometry; however, the expression pattern of CD44s and CD44v was different in Western blot analysis (Figure 4). Furthermore, we detected a 48 kDa band in KYSE70 cells, which is also blocked by the CD44 peptide (Figure 4D). By the radio labeled CD44s pulse chasing assay, a CD44s precursor form (about 48 kDa) was first translated, after which the precursor received *N*− and *O*−glycosylation and reached 85 kDa mature form [22]. This result suggests that the 48 kDa band is a non−glycosylated precursor form of CD44 in KYSE70 cells. We previously determined the epitope of C_44_Mab−46 as _174_−TDDDV−_178_ by enzyme−linked immunosorbent assay using synthetic peptides [21]. Although the Thr174 of CD44 have been confirmed as an *O*−glycan site [23], C_44_Mab−46 could recognize the epitope in the absence of glycosylation. In the future, it should be determined whether the 48 kDa CD44 is exposed on cell surface, and *O*−glycosylation at Thr174 affect the recognition by C_44_Mab−46. In KYSE770 cells, C_44_Mab−46 could recognize the triple bands, suggesting the expression of variant isoforms of CD44. Compared to previously established CD44 mAb C_44_Mab−5, C_44_Mab−46 tended to show the high reactivity to detect endogenous CD44 in Western blotting.

CD44s and CD44v interact with various proliferation−, migration− and invasion−promoting membrane proteins. CD44 binds to platelet−derived growth factor β−receptor and transforming growth factor−β type I receptor [24], and modulates their functions. CD44s also interacts with podoplanin (PDPN) and colocalizes at cell−surface protrusions. PDPN−induced migration requires CD44 in MDCK cells, and knockdown of CD44 and PDPN in oral SCC cells affect cell spreading, suggesting that CD44 directly interacts with PDPN and modulates the intracellular signaling and cell migration [25]. It is interesting that C_44_Mab−46 inhibits these protein–protein interactions and influences their functions.

CD44v6 is highly expressed in many cancers, and acts as co−receptor for at least three receptor tyrosine kinases (c−Met [26,27], EGFR [28], and VEGFR−2 [29]), which contribute to the oncogenic functions of CD44v6. Therefore, humanized anti−CD44v6 mAb BIWA4 (bivatuzumab)−mertansine drug conjugate, was evaluated in clinical trials [30]. However, the clinical trials were discontinued because of the severe skin toxicities [31,32]. A first−in−human phase I clinical trial with RG7356, targeting to the constant region of CD44, showed an acceptable safety profile in patients with advanced CD44−expressing solid tumors. However, the study was terminated due to no evidence of a clinical and/or pharmacodynamic dose–response relationship with RG7356, but not due to safety concerns [33]. Therefore, the development of anti−CD44 targeting mAbs with more potent and fewer side effects is desired.

We previously established cancer−specific mAbs (CasMabs) targeting to PDPN [34,35,36,37] and podocalyxin [38], which are expressed on many cancers, including ESCC [39,40,41,42]. These CasMabs recognize cancer−specific aberrant glycosylation of the target proteins [43]. It is worthwhile to develop a cancer−specific anti−CD44 antibody using the CasMab method. We used the glioblastoma LN229−secreted CD44v3−10 ectodomain as an antigen and generated many mAbs other than C_44_Mab−46. We selected the mAbs, which react with cancer cells, but not with normal cells by flow cytometry and immunohistochemistry. Furthermore, we previously converted the IgG_1_ subclass of C_44_Mab−5 into a mouse IgG_2a_, named 5−mG_2a_, and produced a defucosylated version, 5−mG_2a_−f. The 5−mG_2a_−f exhibited ADCC/CDC activities in vitro and significantly reduced tumor growth in oral cancer cells [44]. Therefore, the production of a class switched and defucosylated version of C_44_Mab−46 is warranted to evaluate the in vivo anti−tumor activity. Since KYSE70 was used in this study, it was reported to be tumorigenic in athymic nude mice [45]; the cell line can be used to evaluate the anti−tumor activity as in the in vivo model.

## 4. Materials and Methods

### 4.1. Cell Lines

Mouse multiple myeloma P3X63Ag8U.1 (P3U1), a glioblastoma cell line (LN229) [46], and CHO−K1 cell lines were obtained from the American Type Culture Collection (ATCC, Manassas, VA, USA). Esophageal squamous cell carcinoma (ESCC) cell lines, KYSE70 and KYSE770 were obtained from the Japanese Collection of Research Bioresources (Osaka, Japan).

Human CD44s ORF was amplified from LN229 cDNA using a HotStar HiFidelity Polymerase Kit (Qiagen Inc., Hilden, Germany). CD44s ORF was subcloned into pCAG−Ble−ssPA16 vector possessing signal sequence and N−terminal PA16 tag (GLEGGVAMPGAEDDVV) [19,47,48,49,50,51]. CHO/CD44s was established by transfecting pCAG−Ble/PA16−CD44s into CHO−K1 cells using a Neon transfection system (Thermo Fisher Scientific, Inc., Waltham, MA, USA). CD44v3−10 ectodomain (CD44ec)−secreting LN229 (LN229/CD44ec) was established by transfecting pCAG−Neo/PA−CD44ec−RAP−MAP into LN229 cells using the Neon transfection system. The amino acid sequences of the tag system in this study were as follows: PA tag [52,53,54], 12 amino acids (GVAMPGAEDDVV); RAP tag [55,56], 12 amino acids (DMVNPGLEDRIE); and MAP tag [57,58], 12 amino acids (GDGMVPPGIEDK).

LN229, KYSE70, and KYSE770 were cultured in Dulbecco’s Modified Eagle Medium (DMEM) complete medium, containing DMEM (4.5 g/L glucose) with L−Gln and without sodium pyruvate (Nacalai Tesque, Inc., Kyoto, Japan), 10% (*v*/*v*) heat−inactivated fetal bovine serum (FBS, Thermo Fisher Scientific Inc.), 100 U/mL of penicillin (Nacalai Tesque, Inc.), 100 μg/mL streptomycin (Nacalai Tesque, Inc.), 0.25 μg/mL amphotericin B (Nacalai Tesque, Inc.). LN229/CD44ec was cultured in DMEM complete medium, containing 0.5 mg/mL of G418 (Nacalai Tesque, Inc.).

P3U1 and CHO−K1 were cultured in Roswell Park Memorial Institute (RPMI)−1640 medium (Nacalai Tesque, Inc.) supplemented with 10% (*v*/*v*) FBS, 100 U/mL penicillin, 100 μg/mL streptomycin, and 0.25 μg/mL amphotericin B. CHO/CD44s was cultured in RPMI complete medium, including 0.5 mg/mL Zeocin (InvivoGen, San Diego, CA, USA). All cells were grown in a humidified incubator at 37 °C with 5% CO_2_.

### 4.2. Purification of CD44ec

After LN229/CD44ec was cultured using DMEM complete medium without G418, CD44ec was purified from the supernatants using RAP tag system, comprised of an anti−RAP tag mAb (clone PMab−2) and a RAP peptide (GDDMVNPGLEDRIE) [55,56]. The filtered culture supernatant (5 L) was passed through PMab−2−Sepharose (2 mL bed volume), and the same process was repeated three times. The beads were then washed with 100 mL of phosphate−buffered saline (PBS, Nacalai Tesque, Inc.), and eluted with 0.1 mg/mL of RAP peptide in a step−wise manner (2 mL × 10).

### 4.3. Hybridoma Production

Female BALB/c mice (6−weeks old) were purchased from CLEA Japan (Tokyo, Japan). The animals were housed under specific pathogen−free conditions. The Animal Care and Use Committee of Tohoku University approved all animal experiments.

A BALB/c mouse was immunized with CD44ec (150 μg) intraperitoneally (i.p.) with Imject Alum (Thermo Fisher Scientific Inc.). The procedure included three additional immunizations with CD44ec (2nd immunization, 150 μg; 3rd immunization, 100 μg; 4th immunization, 50 μg) followed by a final booster injection of CD44ec (50 μg) two days prior to the harvest of splenic cells.

Subsequently, splenic cells were fused with P3U1 cells using polyethylene glycol 1500 (PEG1500; Roche Diagnostics, Indianapolis, IN, USA). The hybridomas were then grown in RPMI media supplemented with hypoxanthine, aminopterin, and thymidine (HAT) for selection (Thermo Fisher Scientific Inc.). The culture supernatants were screened for the anti−CD44ec antibody production using enzyme−linked immunosorbent assay (ELISA).

### 4.4. ELISA

CD44ec was immobilized on Nunc Maxisorp 96−well immunoplates (Thermo Fisher Scientific Inc.) at a concentration of 1 µg/mL for 30 min at 37 °C. After washing with PBS containing 0.05% (*v/**v*) Tween 20 (PBST; Nacalai Tesque, Inc.), wells were blocked with 1% (*w/**v*) bovine serum albumin (BSA)−containing PBST for 30 min at 37 °C. Culture supernatants were added to each well, followed by peroxidase−conjugated anti−mouse immunoglobulins (1:2000 diluted; Agilent Technologies Inc., Santa Clara, CA, USA). Enzymatic reactions were conducted using 1 Step Ultra TMB (Thermo Fisher Scientific Inc.) followed by the measurement of the optical density at 655 nm, using an iMark microplate reader (Bio−Rad Laboratories, Inc., Berkeley, CA, USA).

### 4.5. Flow Cytometry

Exponentially growing cells were collected following a brief exposure to 0.25% (*w*/*v*) trypsin and 1 mM ethylenediaminetetraacetic acid (Nacalai Tesque, Inc.). The cells were then washed with 0.1% (*w*/*v*) BSA in PBS and treated with C_44_Mab−46 for 30 min at 4 °C. After incubation, cells were treated with Alexa Fluor 488−conjugated anti−mouse IgG (1:2000; Cell Signaling Technology, Inc., Danvers, MA, USA). In the peptide blocking assay, C_44_Mab−46 (1 μg/mL) was incubated with 10 μg/mL of human CD44 peptide (amino acids 161−180, WT) or the D175A mutant peptide [21] for 30 min at 4 °C. CHO/CD44s cells were treated with C_44_Mab−46 + each peptide, and further treated with Alexa Fluor 488−conjugated anti−mouse IgG. Fluorescence data were collected using SA3800 Cell Analyzer (Sony Corp.) and analyzed using FlowJo software (BD Biosciences, Franklin Lakes, NJ, USA).

### 4.6. Determination of Dissociation Constant (K_D_) by Flow Cytometry

CHO/CD44s, KYSE70, and KYSE770 cells were suspended in 100 μL serially−diluted anti−C_44_Mab−46 mAbs, after which 50 μL Alexa Fluor 488−conjugated anti−mouse IgG (1:200; Cell Signaling Technology, Inc.) was added. Fluorescence data were collected, using EC800 (Sony Corp., CHO/CD44s) or BD FACSLyric (KYSE70 and 770). The apparent *K*_D_ was calculated by fitting saturation binding curves to the built−in, one−site binding models in GraphPad PRISM 8 (GraphPad Software, Inc., La Jolla, CA, USA).

### 4.7. Western Blot Analysis

Cell lysates (10 μg) were boiled in sodium dodecyl sulfate (SDS) sample buffer (Nacalai Tesque, Inc.). Proteins were separated on 5–20% polyacrylamide gels (FUJIFILM Wako Pure Chemical Corporation) and transferred onto polyvinylidene difluoride (PVDF) membranes (Merck KGaA, Darmstadt, Germany). After blocking with 4% (*w*/*v*) skim milk (Nacalai Tesque, Inc.) in PBS with 0.05% (*v*/*v*) Tween 20, membranes were incubated with 1 μg/mL C_44_Mab−46 or 1 μg/mL anti−β−actin (clone AC−15; Sigma–Aldrich Corp., St. Louis, MO, USA). Membranes were then incubated with peroxidase−conjugated anti−mouse immunoglobulins (diluted 1:1000; Agilent Technologies, Inc.) to detect C_44_Mab−46 and anti−β−actin. Finally, protein bands were detected with a chemiluminescence reagent, ImmunoStar LD (FUJIFILM Wako Pure Chemical Corporation) using a Sayaca−Imager (DRC Co. Ltd., Tokyo, Japan).

### 4.8. Immunohistochemical Analysis

Paraffin−embedded ESCC tissue microarray (Product Code: BC02011, US Biomax Inc., Rockville, MD, USA) were deparaffinized in xylene and rehydrated. Then, they were autoclaved in citrate buffer (pH 6.0; Agilent Technologies Inc.) for 20 min. After blocking with SuperBlock T20 (Thermo Fisher Scientific, Inc.), sections were incubated with C_44_Mab−46 (5 μg/mL) for 1h at room temperature and then treated with the EnVision+ Kit for mouse (Agilent Technologies Inc.) for 30 min. Color was developed using 3,3′−diaminobenzidine tetrahydrochloride (DAB; Agilent Technologies Inc.) for 2 min. Counterstaining was performed with hematoxylin (FUJIFILM Wako Pure Chemical Corporation). Hematoxylin and eosin (HE) staining (FUJIFILM Wako Pure Chemical Corporation) was performed using consecutive tissue sections. Leica DMD108 (Leica Microsystems GmbH, Wetzlar, Germany) was used to examine the sections and obtain images.

## Figures and Tables

**Figure 1 ijms-23-05535-f001:**
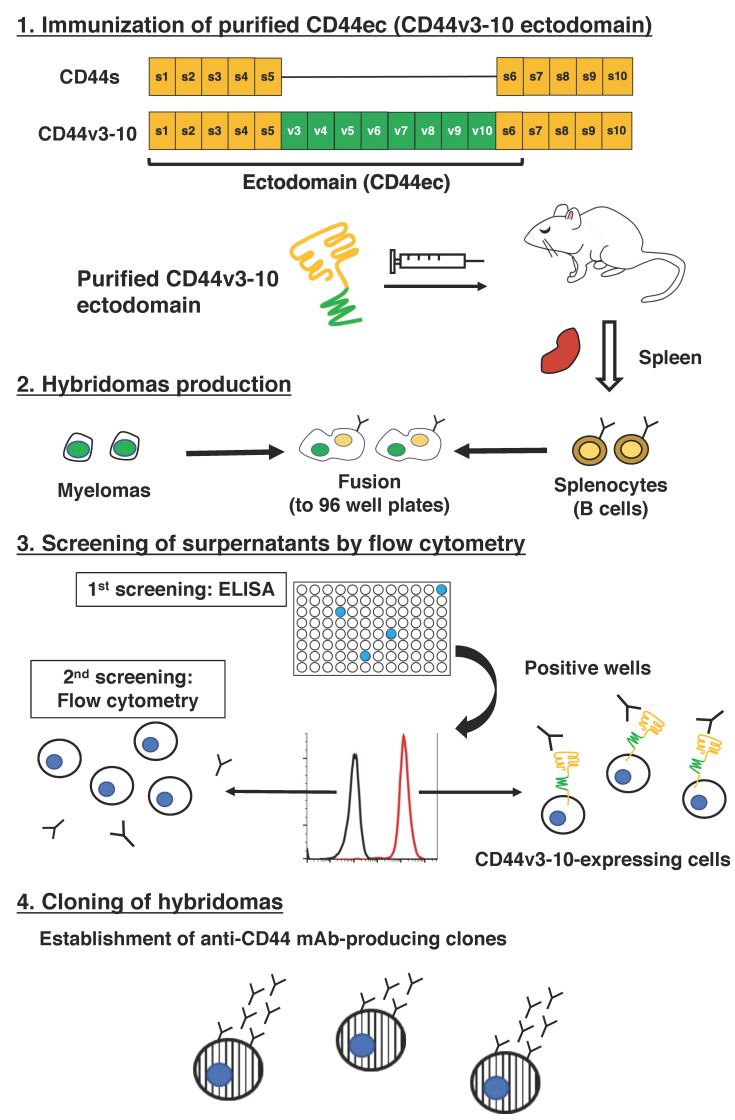
A schematic procedure of anti−human CD44 mAbs production. A BALB/c mouse was intraperitoneally immunized with the purified CD44v3−10 ectodomain. The screening was then conducted by ELISA using the purified CD44v3−10 ectodomain, flow cytometry using parental and CD44v3−10−overexpressed CHO−K1 cells.

**Figure 2 ijms-23-05535-f002:**
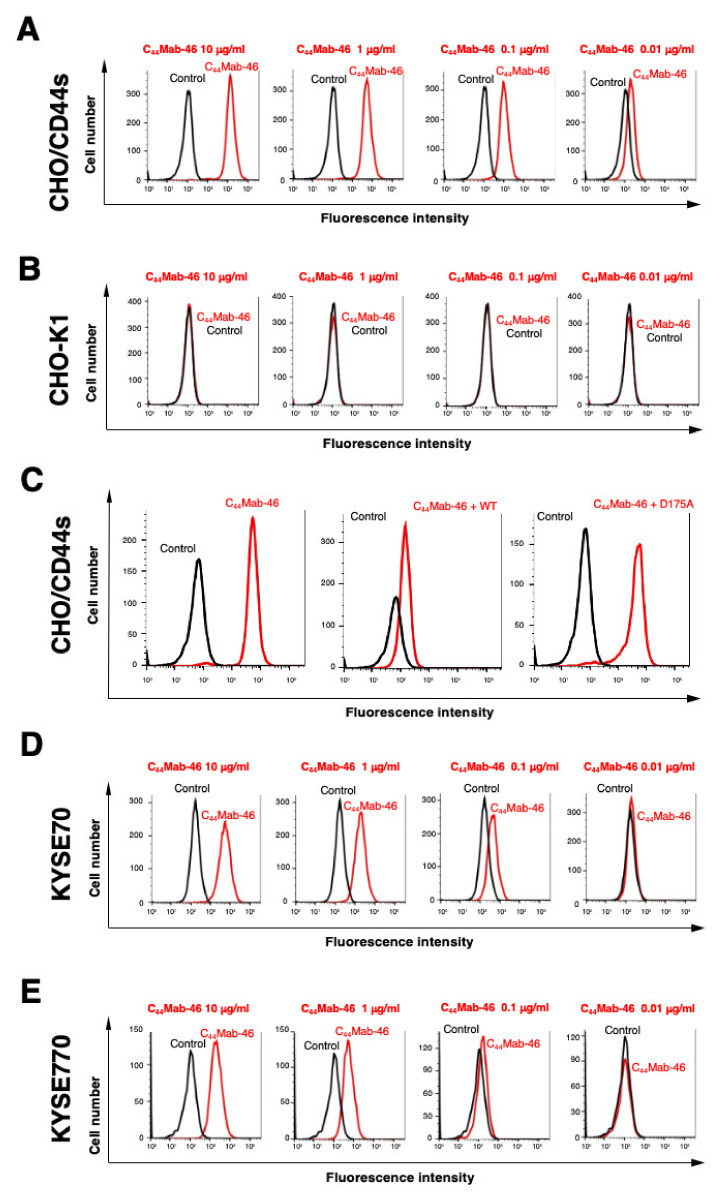
Flow cytometry to CD44 expressing cells using C_44_Mab−46. (**A**,**B**) CHO/CD44s (**A**) and CHO−K1 (**B**) cells were treated with 0.01–10 µg/mL of C_44_Mab−46, followed by treatment with Alexa Fluor 488−conjugated anti−mouse IgG. (**C**) C_44_Mab−46 (1 μg/mL), C_44_Mab−46 (1 μg/mL) plus human CD44 peptide (161−180 amino acids, WT) or alanine−substituted peptide (D175A) (10 μg/mL), and control (blocking buffer) were reacted with CHO/CD44 cells for 30 min at 4 °C, followed by treatment with Alexa Fluor 488−conjugated anti−mouse IgG. (**D**,**E**) KYSE70 (**D**) and KYSE770 (**E**) cells were treated with 0.01–10 µg/mL of C_44_Mab−46, followed by treatment with Alexa Fluor 488−conjugated anti−mouse IgG. The black line represents the negative control (blocking buffer).

**Figure 3 ijms-23-05535-f003:**
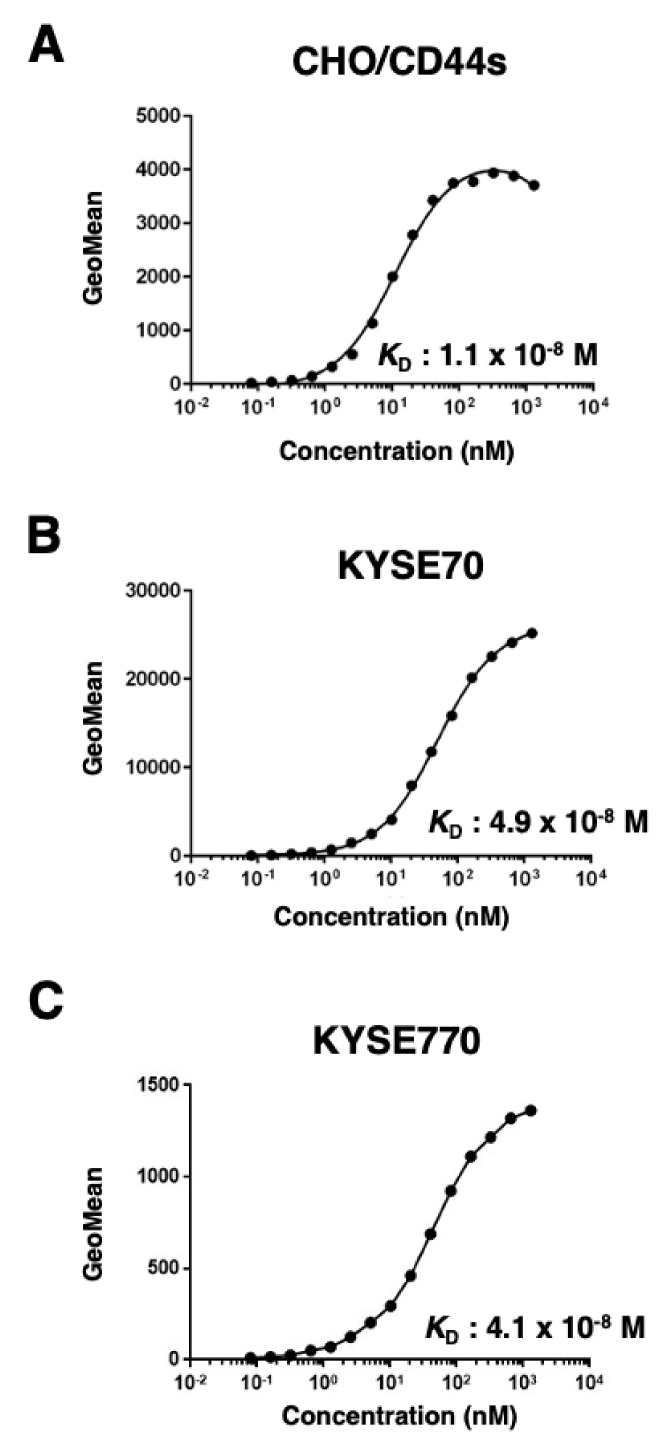
The determination of the binding affinity of C_44_Mab−46. CHO/CD44s (**A**), KYSE70 (**B**), and KYSE770 (**C**) cells were suspended in 100 µL serially diluted C_44_Mab−46 (0.08 to 1300 nM). Then, cells were treated with Alexa Fluor 488−conjugated anti−mouse IgG. Fluorescence data were subsequently collected, followed by the calculation of the apparent dissociation constant (*K*_D_) by GraphPad PRISM 8.

**Figure 4 ijms-23-05535-f004:**
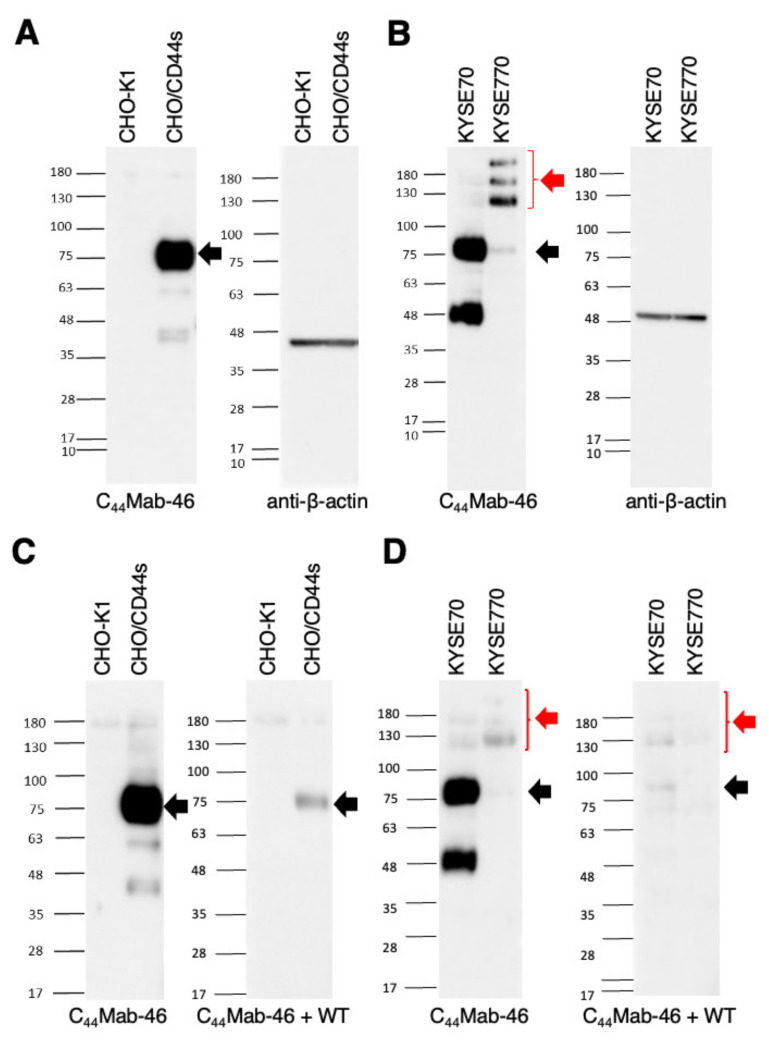
Western blotting by C_44_Mab−46. (**A**) Cell lysates of CHO−K1 and CHO/CD44s (10 µg) were electrophoresed and transferred onto polyvinylidene fluoride (PVDF) membranes. The membranes were incubated with 1 µg/mL of C_44_Mab−46 and 1 µg/mL of anti−β−actin and subsequently with peroxidase−conjugated anti−mouse immunoglobulins. (**B**) Cell lysates of KYSE70 and KYSE770 (10 µg) were electrophoresed and transferred onto PVDF membranes. The membranes were incubated with 1 µg/mL of C_44_Mab−46 and 1 µg/mL of anti−β−actin and subsequently with peroxidase−conjugated anti−mouse immunoglobulins. (**C**) Cell lysates of CHO−K1 and CHO/CD44s (10 µg) were electrophoresed and transferred onto PVDF membranes. The membranes were incubated with 1 µg/mL of C_44_Mab−46 and C_44_Mab−46 (1 μg/mL) plus the CD44 peptide (10 μg/mL, WT), and subsequently with peroxidase−conjugated anti−mouse immunoglobulins. (**D**) Cell lysates of KYSE70 and KYSE770 (10 µg) were electrophoresed and transferred onto PVDF membranes. The membranes were incubated with 1 µg/mL of C_44_Mab−46 and C_44_Mab−46 (1 μg/mL) plus the CD44 peptide (10 μg/mL, WT), and subsequently with peroxidase−conjugated anti−mouse immunoglobulins. Black arrows indicate the predicted size of CD44s (~85 kDa). The red arrow indicates the CD44 variants.

**Figure 5 ijms-23-05535-f005:**
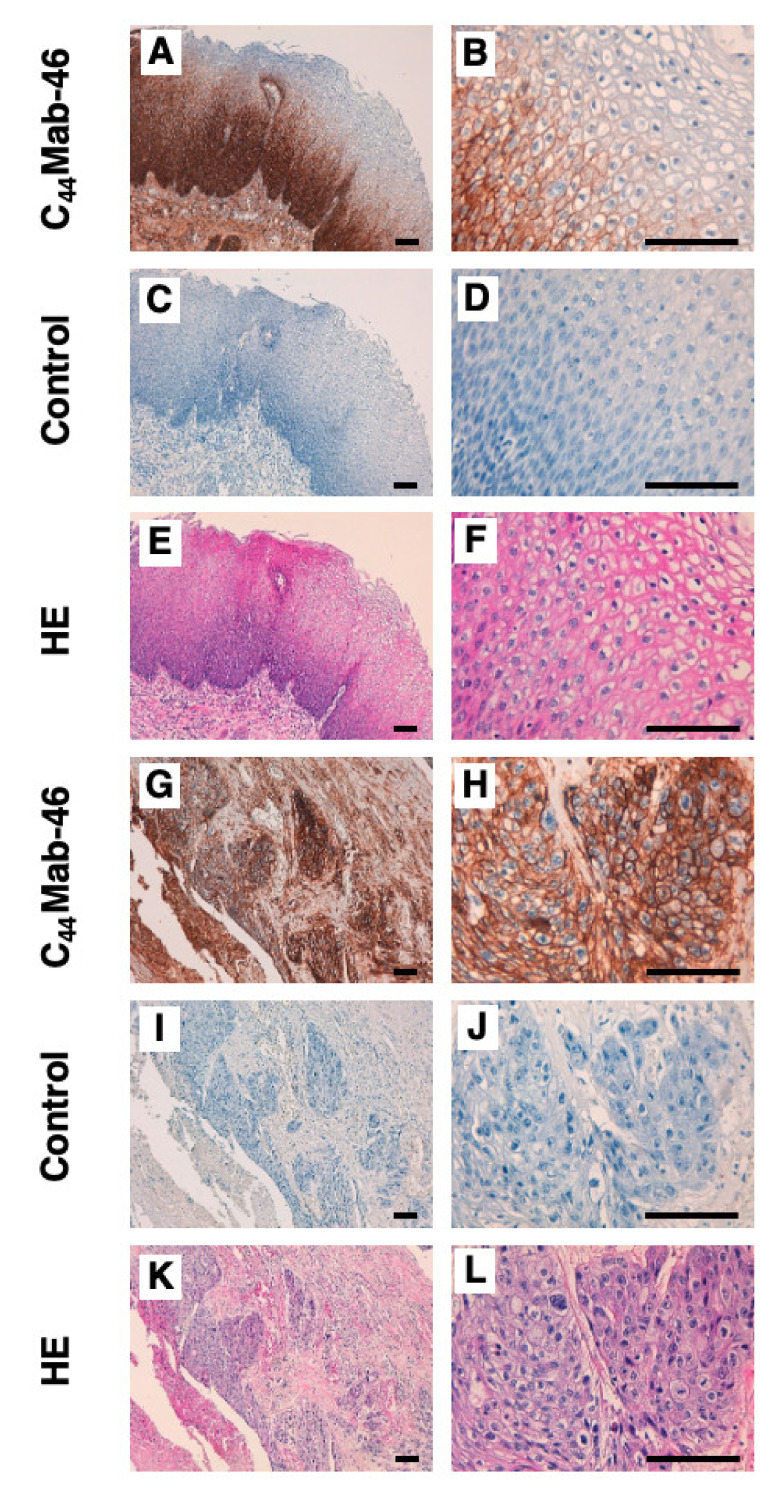
Immunohistochemical analysis using C_44_Mab−46 against ESCC tissues. (**A**–**D**,**G**–**J**) After antigen retrieval, sections were incubated with 5 µg/mL of C_44_Mab−46 or negative control (blocking buffer) followed by treatment with the Envision+ kit. Color was developed using DAB, and sections were counterstained with hematoxylin. (**E**,**F**,**K**,**L**) Hematoxylin and eosin (HE) staining. Scale bar = 100 µm.

**Table 1 ijms-23-05535-t001:** Immunohistochemical analysis using C_44_Mab−46 against ESCC tissues.

Case No.	Age	Sex	Organ/Anatomic Site	Pathology diagnosis	Grade	Type	Intensity
1	52	M	Esophagus	Squamous cell carcinoma	I	malignant	++
2	62	F	Esophagus	Squamous cell carcinoma	I	malignant	++
3	54	M	Esophagus	Squamous cell carcinoma	I	malignant	++
4	59	M	Esophagus	Squamous cell carcinoma	I	malignant	+
5	60	M	Esophagus	Squamous cell carcinoma	II	malignant	++
6	66	M	Esophagus	Squamous cell carcinoma	−	malignant	+
7	36	M	Esophagus	Squamous cell carcinoma	I	malignant	++
8	55	F	Esophagus	Squamous cell carcinoma	I	malignant	−
9	59	F	Esophagus	Squamous cell carcinoma	I − II	malignant	++
10	48	M	Esophagus	Squamous cell carcinoma	I − II	malignant	++
11	41	M	Esophagus	Squamous cell carcinoma	I	malignant	++
12	58	M	Esophagus	Squamous cell carcinoma	I	malignant	++
13	56	M	Esophagus	Squamous cell carcinoma	I	malignant	++
14 (Figure 5G–L)	72	F	Esophagus	Squamous cell carcinoma	I	malignant	+++
15	41	M	Esophagus	Squamous cell carcinoma	I − II	malignant	+++
16	50	M	Esophagus	Squamous cell carcinoma	I	malignant	−
17	48	M	Esophagus	Squamous cell carcinoma	I	malignant	+
18	55	M	Esophagus	Squamous cell carcinoma	I − II	malignant	−
19	61	F	Esophagus	Squamous cell carcinoma	II	malignant	+
20	35	M	Esophagus	Squamous cell carcinoma	I	malignant	++
21	72	M	Esophagus	Squamous cell carcinoma	I	malignant	++
22	70	M	Esophagus	Squamous cell carcinoma	−	malignant	+++
23	42	F	Esophagus	Squamous cell carcinoma	I	malignant	++
24	53	M	Esophagus	Squamous cell carcinoma	I	malignant	++
25	54	M	Esophagus	Squamous cell carcinoma	I − II	malignant	++
26 (Figure 5A–F)	54	F	Esophagus	Squamous cell carcinoma	I − II	malignant	+++
27	65	F	Esophagus	Squamous cell carcinoma	I − II	malignant	++
28	63	F	Esophagus	Squamous cell carcinoma	I − II	malignant	+++
29	62	M	Esophagus	Squamous cell carcinoma	II	malignant	+++
30	63	M	Esophagus	Squamous cell carcinoma	II	malignant	++
31	65	M	Esophagus	Squamous cell carcinoma	II	malignant	+++
32	64	F	Esophagus	Squamous cell carcinoma	II	malignant	++
33	71	M	Esophagus	Squamous cell carcinoma	II	malignant	+++
34	55	M	Esophagus	Squamous cell carcinoma	II	malignant	+++
35	57	F	Esophagus	Squamous cell carcinoma	II	malignant	+++
36	56	M	Esophagus	Squamous cell carcinoma	II	malignant	+++
37	60	M	Esophagus	Squamous cell carcinoma	II − III	malignant	+++
38	61	F	Esophagus	Squamous cell carcinoma	II	malignant	++
39	61	M	Esophagus	Squamous cell carcinoma	II − III	malignant	+++
40	50	M	Esophagus	Smooth muscle and fatty tissue	−	malignant	+++
41	66	M	Esophagus	Squamous cell carcinoma	II	malignant	+++
42	45	M	Esophagus	Squamous cell carcinoma	II	malignant	++
43	68	M	Esophagus	Squamous cell carcinoma	II	malignant	+++
44	58	M	Esophagus	Squamous cell carcinoma	II	malignant	+++
45	57	M	Esophagus	Squamous cell carcinoma	II − III	malignant	++
46	54	F	Esophagus	Squamous cell carcinoma	II	malignant	+
47	48	M	Esophagus	Squamous cell carcinoma	II	malignant	++
48	68	M	Esophagus	Squamous cell carcinoma	II	malignant	+++
49	54	M	Esophagus	Squamous cell carcinoma	II	malignant	+++
50	70	M	Esophagus	Squamous cell carcinoma	III	malignant	+++
51	72	M	Esophagus	Squamous cell carcinoma	III	malignant	++
52	62	M	Esophagus	Squamous cell carcinoma	III	malignant	−
53	63	M	Esophagus	Squamous cell carcinoma	II − III	malignant	++
54	49	F	Esophagus	Squamous cell carcinoma	II − III	malignant	++
55	53	F	Esophagus	Squamous cell carcinoma	II − III	malignant	++
56	61	M	Esophagus	Squamous cell carcinoma	III	malignant	+++
57	61	M	Esophagus	Squamous cell carcinoma	II	malignant	+++
58	59	F	Esophagus	Squamous cell carcinoma	III	malignant	+++
59	62	M	Esophagus	Squamous cell carcinoma	II − III	malignant	+++
60	56	M	Esophagus	Squamous cell carcinoma	III	malignant	+++
61	73	F	Esophagus	Squamous cell carcinoma	II − III	malignant	+++
62	57	M	Esophagus	Squamous cell carcinoma	II − III	malignant	++
63	64	M	Esophagus	Squamous cell carcinoma	III	malignant	+++
64	60	M	Esophagus	Squamous cell carcinoma	II − III	malignant	+++
65	66	M	Esophagus	Squamous cell carcinoma	III	malignant	+++
66	67	M	Esophagus	Squamous cell carcinoma	II − III	malignant	+++
67	75	M	Esophagus	Squamous cell carcinoma	III	malignant	+++

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
