# Peer review of "Development of a Novel Anti−CD44 Monoclonal Antibody for Multiple Applications against Esophageal Squamous Cell Carcinomas"

_ijms, 2022, doi:10.3390/ijms23105535_

Round 1

Reviewer 1 Report

This manuscript discusses the generation of anti-CD44 antibodies. These antibodies were generated through the immunization of BLAB/c mice using CD44 ectodomain CD44v3-10 (CD44ec) isoform. A mouse splenic cells were isolated and fused to myeloma cells generating a hybridoma that produces antibodies against CD44s/CD44ec into the culture medium. Screening for antibodies was carried out using enzyme-linked immunosorbent assay (ELISA) by immobilizing the CD44ec on the 96-well plate and C44Mab-46 antibody was selected. The antibody, C44Mab-46, was characterized through western blotting to test for the antibody sensitivity, which showed it to bind to the CD44 standard and CD44 isoform. The dissociation constant of C44Mab-46 was also determined though flow cytometry using CHO/CD44s as well as KYSE70 and KYSE770 cells which expressed CD44 standard and CD44 isoform. Recall that  CD44 is encoded by a gene composed of 19 exons which undergo alternative splicing events generating various CD44 isoforms. CD44 is expressed in various tissues throughout the human body, but is heavily expressed in cancer cells and is considered a biomarker for cancer diagnoses as well as therapy.

This paper is well written and straightforward. The goal, objective, and conclusion drawn are clear and well explained as well, which is generating a novel antibody(C44-Mab-46) that has the ability to bind CD44 glycoprotein produced by cancer cells, which can be used as a tool to investigate the presence of CD44 in various applications. I enjoyed reading this manuscript as it will be a step further in generating therapeutics against cancer.

The manuscript can be improved by carrying

 Minor revisions:

  • The first sentence of the abstract can be revised and instead of saying that CD44 is heavily expressed in normal and cancer cells, maybe it can be edited that CD44 is heavily expressed in cancer cells.
  • In the introduction section, more background on CD44 is needed such as how many isoforms of CD44 and which tissues express CD44 glycoprotein. Also, CD44s and CD44v isoforms share overlapping as well as distinct functional roles. In addition, CD44 isoforms exhibit additional binding motifs promoting the interaction of CD44 with molecules in the microenvironment and can “act as co-receptors by sequestering growth factors on the cell surface presenting these to their specific receptors”. With the limited knowledge about CD44, it first the CD44 need to be investigated more in depth to define the functional roles of the CD44 isoforms in cancer and to determine the potential benefits of targeting these CD44 isoforms or their signaling pathways for cancer therapy.
  • Maybe it would be good to add a schematic figure of CD44 in the introduction
  • I was confused when it mentioned that the antibody C44Mab-44 was also able to stain normal squamous epithelium of esophagus in the immunohistochemical analysis experiment. Does this mean the mAb interacts with healthy cells? This should be discussed
  • Carryout Western blotting to detect for antibodies and use a negative control, for example, an antibody that is known to not bind CD44.
  • Use another cell line which it is known to not express any of the CD44 isoforms as a negative control when determining the dissociation constant.
  • Need to include in the cell confluency or concentration which was used in each experiment.
  • Do western blotting to detect if the antibody in the culture supernatant is probable to test if antibodies recognize CD44ec in case it has various conformational epitopes. in addition to using CD44s not only CD44ec. Also, it was not mentioned in the materials and methods which cell line was used in this experiment
  • Also, would it be good to do CD44s not only CD44ec to check for the affinity of the C44-Mab-46 to CD44s unless they have a conserved epitope.
  • For the flow cytometry experiments, it was mentioned that “Cells were collected” but did not identify which cells were being discussed nor the cell density/concentration of cells used for this experiment.
  • For determining the KD by flow cytometry, use cells that you know don’t express CD44s/CD44ec as a negative control. Or maybe do SPR experiments to determine the Kd but instead of cells the CD44 can be used.
  • Peptides test, maybe another binding experiment to check the binding ability of the 174-TDDDV-178 vs the mutant. So, does that mean that the antibody is able to interact with one residue?

Reviewer 2 Report

In this work, the authors reported interesting studies on “Development of a novel anti-CD44 monoclonal antibody for multiple applications against esophageal squamous cell carcinomas". They have established anti-CD44 monoclonal antibodies (mAbs) by immunizing mice with a CD44 variant (CD44v3-10) ectodomain and screening using enzyme-linked immunosorbent assay. Finally, demonstrating C44Mab-46 is very useful for detecting CD44 in various applications. Generally speaking, the idea of the paper was clearly stated, detailed studies on the preparation, purification, ELISA assay, dissociation constant measurement …. were well presented, The reviewer believe that this is a very interesting article with novelty. In addition, the idea of the paper was clearly stated and the structure was well organized.

This paper can be accepted after a minor revision.

  1. Error bars should be added in the Figures.

Round 2

Reviewer 1 Report

The authors have addressed all of my concerns.